# The miR-183 Cluster: Biogenesis, Functions, and Cell Communication via Exosomes in Cancer

**DOI:** 10.3390/cells12091315

**Published:** 2023-05-05

**Authors:** Shuhui Li, Wei Meng, Ziyi Guo, Min Liu, Yanyun He, Yanli Li, Zhongliang Ma

**Affiliations:** 1Lab for Noncoding RNA & Cancer, School of Life Sciences, Shanghai University, Shanghai 200444, China; 2Experimental Center of Life Sciences, Shanghai University, Shanghai 200444, China

**Keywords:** miR-183 cluster, noncoding RNAs, exosome, cancer

## Abstract

Cancer is one of the leading causes of human death. MicroRNAs have been found to be closely associated with cancer. The miR-183 cluster, comprising miR-183, miR-96, and miR-182, is transcribed as a polycistronic miRNA cluster. Importantly, in most cases, these clusters promote cancer development through different pathways. Exosomes, as extracellular vesicles, play an important role in cellular communication and the regulation of the tissue microenvironment. Interestingly, the miR-183 cluster can be detected in exosomes and plays a functional regulatory role in tumor development. Here, the biogenesis and functions of the miR-183 cluster in highly prevalent cancers and their relationship with other non-coding RNAs are summarized. In addition, the miR-183 cluster in exosomes has also been discussed. Finally, we discuss the miR-183 cluster as a promising target for cancer therapy. This review is expected to provide a new direction for cancer treatment.

## 1. Introduction

Cancer is the world’s largest public health problem and one of the main causes of human death [1]. Worldwide, an estimated 19.3 million new cancer cases (18.1 million excluding nonmelanoma skin cancer) and almost 10.0 million cancer deaths (9.9 million excluding nonmelanoma skin cancer) occurred in 2020 [2]. In China, the mortality rate of cancer is relatively high and has been on the rise over recent years, among which, lung cancer ranks first [3,4,5]. In addition to lung cancer, colorectal cancer (CRC), prostate cancer (PCa), gastric cancer (GC), and breast cancer are all high-incidence cancers [6,7,8,9]. Malignant tumors have the characteristics of unrestricted proliferation, invasion, and migration [10]. At present, the main methods for cancer treatment include surgery, chemotherapy, radiotherapy, and targeted drug therapy. The improvement of these treatment models is required from a wider and more effective perspective, because the results of the current treatment models fall short of expectations. Most cancers are not obvious in their early stages and are highly invasive. When diagnosed, they are already at an advanced stage, causing patients to miss out on optimal treatment and survival rates to be extremely low [11,12,13,14,15]. Therefore, the early screening and diagnosis of cancer are of great importance for this treatment. Interestingly, it has been shown that non-coding RNAs (ncRNAs) play a key role in regulating the development and progression of various diseases, especially cancer [16,17]. It is ncRNAs that have a high potential to become a new strategy for its clinical diagnosis and treatment.

MicroRNAs (miRNAs) are a class of single-stranded RNAs that are 18–23 nucleotides in length and do not encode proteins [18]. They function in the post-transcriptional regulation of gene expression and are powerful regulators of various cellular activities, including cell growth, differentiation, development, and apoptosis [19]. It is thought that up to 60% of human protein-coding genes may be regulated by miRNAs and that each miRNA is capable of regulating the expression of multiple target genes [20]. Importantly, they are closely associated with the occurrence, development, invasion, and migration of different types of cancer [21,22].

The miR-183 cluster consists of three members: miR-183, miR-96, and miR-182. Reports have confirmed that the miR-183 cluster has regulatory roles in cancer, autoimmune diseases, neuronal diseases, psychiatric diseases, and other human diseases [23]. In addition, miR-183 cluster expression can be regulated by other ncRNAs to influence its downstream target genes to achieve cancer-promoting or cancer-suppressing functions [24]. In addition, it can be packaged in exosomes, which allows it to be found in the exosomes of human serum and transferred between cells using a microencapsulation-dependent mechanism [25,26,27]. In this paper, we discuss the biogenesis of the miR-183 cluster, summarize its regulatory role in various cancer species, and evaluate the application of miR-183 clusters as a novel diagnostic method for cancer therapy.

## 2. Biogenesis of miR-183 Cluster

Of these three cluster members, miR-96 was discovered first in the HeLa cancer cell line, and miR-183 and miR-182 were found in 2003 [23]. According to the miRBase search, the relative position of the miR-183 cluster on the chromosome is highly conserved across animals. The miR-96 gene is in the middle and the miR-182 and miR-183 genes are located either side. The miR-183 cluster distribution on the genomes of different species is shown [28,29] (Figure 1A). The sequences among the members of the miR-183 cluster show some similarity. The mature sequences of three members of the miR-183 cluster from humans, mice, zebrafish, and amphioxus are compared (Figure 1B). The results show that six bases in the nucleotide sequence of the first eight positions of the 5′ end of the miR-183/miR-96/miR-182 mature sequence are identical. Furthermore, seven bases in the nucleotide sequence of the first eight positions of the 5′ end of the miR-183/miR-96 and miR-96/miR-182 mature sequences are identical. The similarity of these seed sequences suggests that the three miRNAs are paralogous and that there is some consistency in the target sites that they recognize. The sequence homology of the miR-183 cluster and their conservation of genome organization as a cluster in bilateral organisms suggest that there is an evolutionary advantage in retaining this miRNA cluster. In addition, the maturation sequences of each miRNA in the miR-183 clusters from several animals are compared (Figure 1C). The miR-183 seed sequence is consistent, except for a change of base from U to A at the 5′ end in *saccoglossus kowalevskii*. The miR-96 seed sequence is 100% similar, and miR-182 is less similar than miR-183 and miR-96.

The miR-183 cluster originating from the human chromosome 7q32.2 was transcribed into primary transcription products via RNA polymerase II, which was named pri-miRNAs. After being digested by the Drosha enzyme, pri-miRNA became a precursor miRNA with a hairpin structure (pre-miRNA). Each of the three members of the miR-183 cluster had its own unique precursor: pre-miR-183(MI0000273), pre-miR-182(MI0000272), and pre-miR-96(MI0000098). After initial shearing, the pre-miRNA was transported from the nucleus to the cytoplasm via the action of the transporter protein exportin-5. The TRBP combined with the Dicer enzyme to produce mature miRNA, which further combined with the Ago 2 protein to form the RISC (RNA-induced silencing complex) complex. It was generally thought that miRNAs regulate their targets either by translation inhibition or by target destabilization (Figure 2). It was further found that the three members of the miR-183 cluster were abnormally expressed in many tumors and activated tumorigenesis.

## 3. Biological Roles and Mechanisms of miR-183 Cluster in Cancer

In recent years, the mechanism of the miR-183 cluster has been unmasked in tumorigenesis. Here, the progress of the miR-183 cluster in cancer is summarized.

### 3.1. miR-183 in Cancer

miR-183 is promising for its use as a specific diagnostic biomarker because it has been found to influence the development of high-incidence cancers and to have a positive correlation with tumor size, proliferative capacity, and migration ability [30,31,32,33,34].

miR-183 could specifically modulate the proliferation and migration of lung cancer cells. Our laboratory demonstrated that miR-183-5p was necessary for the progression of non-small cell lung cancer (NSCLC) by inhibiting the phosphatase and tensin homolog deleted on chromosome 10 (PTEN). Further experiments indicated that miR-183-5p suppressed p53 and activated AKT signaling through phosphorylation [35]. In addition, it could promote the epithelial–mesenchymal transformation (EMT) of H1299 cells, resulting in radiation resistance [36].

Similarly, miR-183 plays a regulatory role in CRC and PC. In CRC, Huang et al. [37] found that the ultraviolet-radiation-resistance-associated gene (UVRAG) was a target of miR-183, a key regulator of promoting autophagy and apoptosis. In PCa, both tropomyosin 1 (TPM1) and high-mobility group nucleosome-binding domain 5 (HMGN5) were target genes of miR-183. Upregulated TPM1 inhibited exosome-derived miR-183 and miR-183-3p downregulated HMGN5 to inhibit the PCa cell growth and metastasis [38,39].

miR-183 has been discovered to be involved with GC cells. Eukaryotic elongation factor 2 (EEF2) [40], UVRAG [41], and TPM1 [42], etc., were all regulated by miR-183, which played a regulatory role in the occurrence and development of GC. In human GC cells, EEF2 restricted GC cell proliferation and migration by upregulating miR-183-5p. MiR-183 suppressed starvation-induced autophagy and death by targeting UVRAG. TPM1-targeted miR-183-5p was able to lower the TPM1 expression in gastric cancer tissues and cell lines in comparison to that in the surrounding healthy tissues and gastric epithelial cells. Interestingly, miR-183-5p was shown to be expressed as different 5′isomiRs with different functions. Three 5′isomiRs of miR-183-5p were highly upregulated in breast cancer. The influence of miR-183-5p|+2 for cell vitality was higher than the influences of miR-183-5p|0 and miR-183-5 p|+1 [43].

In NSCLC, hepatocellular carcinoma (HCC), and osteosarcoma (OS), the regulatory networks of metastasis-associated gene 1 (MTA1) and miR-183 were extremely significant [44,45,46]. The protein expression and mRNAs of MTA1 were increased. MiR-183 inhibited the proliferation, EMT, migration, and invasion of tumor cells by downregulating MTA1.

All the relevant information about miR-183 is summarized (Table 1). These results demonstrate that miR-183 exhibits an abnormal expression in a range of malignancies, indicating that it has a global effect on tumor growth. However, its expression can appear contradictory. For example, in CRC, the relative expression level of miR-183 is upregulated in tissues, while the induction of autophagy leads to the downregulation of the miR-183 in colorectal cancer cells. In PCa, the expression of miR-183 and miR-183-3p in tissues shows opposite results. Overall, miR-183 can achieve cancer promotion through different pathways, which is expected to become a new marker for cancer diagnosis.

### 3.2. miR-182 in Cancer

In lung cancer, the overexpression of miR-182 can significantly promote cell proliferation, migration, and invasion. Studies have shown that forkhead box O3 (FOXO3) [49], EPAS1 [50], homeobox A9 (HOXA9) [51], STARD13 [52], FBXW7, FBXW11 [53], programmed cell death 4 (PDCD4) [54], glioma-associated oncogene homolog 2 (GLI2) [55], and cortactin (CTTN) [56] are miR-182 target genes, which plays an important regulatory role. Specifically, FOXO3, EPAS1, FBXW7, FBXW11, PDCD4 HOXA9, and STARD13 are down-regulated and negatively correlated with miR-182 expression. On the contrary, GLI2 and CTTN are overexpressed and targeted by miR-182. miR-182 is upregulated in lung cancer, acting as an independent prognostic factor for tumor recurrence in patients.

In CRC, miR-182 affects tumor cell development by regulating specific pathways. It inhibits the expression of ST6GALNAC2 and the PI3K/AKT signal is partially blocked due to the change in the ST6GALNAC2 level [6,57]. Lnc-AGER-1 changes the expression of its adjacent gene AGAGER by acting as a competitive endogenous RNA of miR-182 in CRC [58].

miR-182, which is important for tumor development, has also been discovered in PCa. It is abnormally overexpressed and promotes tumor cell development by regulating the ST6GALNAC5/PI3K/AKT axis [59]. PDCD4, as its target gene, is also regulated [60]. MiR-182 inhibits FOXO1 levels to increase PCa proliferation, migration, and invasion [61]. The androgen receptor directly regulates the transcription of miR-182-5p, which can target the 3′UTR of ARRDC3 mRNA and affect the expression of ARRDC3/ITGB4 to promote cancer [62]. Moreover, it can significantly activate the Wnt/β-catenin pathway by targeting multiple of its negative regulators, including glycogen synthase kinase-3beta (GSK-3β), adenomatous polyposis coli (APC), casein kinase 1 (CK1), and Axin [63].

In GC, it has been shown that miR-182 receives circRNA regulation. For instance, the miR-182/MTSS1 axis can be regulated by circ002059 to inhibit GC cell proliferation and migration and the xenograft tumor growth in mice [64]. In addition, silencing circ0001658 inhibits cell viability and inhibits RAB10 expression by sponging miR-182 to promote GC cell apoptosis [65].

In breast cancer, on the one hand, miR-182 targets and regulates the expression of CD3D, IL-2-inducible tyrosine kinase (ITK), FOXO1, nuclear factor of activated T cells (NFATs), TGFβ, and Toll-like receptor 4 (TLR4) to promote progression. On the other hand, a study found that the delivery of miR-182 inhibitors via extracellular vesicle therapy effectively targets macrophages to reduce miR-182 expression. Furthermore, miR-182 also plays a role in bladder cancer. Inhibiting Cofilin 1 expression suppresses bladder tumor cell proliferation, migration, invasion, and colony formation efficiency [66,67,68].

In light of the summarized aforesaid factors (Table 2), miR-182 is aberrantly overexpressed in a variety of cancer types and can promote cancer through a variety of mechanisms. ST6GALNAC2 in CRC and ST6GALNAC5 in PCa are both targets of miR-182. Thus, whether miR-182 can regulate the ST6GALNAC family is speculative. Therefore, we envision whether miR-182 can be used as a marker for early cancer screening as well as a target for specific therapy. Above all, we hope that the analysis of the current status of miR-182 research will provide a theoretical basis for more studies on miR-182 and its clinical application in the future.

### 3.3. miR-96 in Cancer

miR-96 has been reported in a variety of highly prevalent tumors (Table 3). It might be a non-invasive diagnostic and prognostic marker for NSCLC and could affect the migration, invasion, and cisplatin resistance of lung cancer cells by targeting glypican-3 (GPC3) and SAMD9 [70,71,72]. In CRC, miR-96 delays carcinogenesis through the AMPKα2/FTO/m6A/MYC [73] network, whereas an interaction with the reversion-inducing cysteine rich protein with Kazal motifs (RECK) [74] promotes invasion. Interestingly, miR-96 affects PCa, GC, and HCC by regulating the expression of FOXO1 and FOXO3a [75,76,77,78,79]. In addition, it has been pointed out that miR-96 interacts with MTSS1 [80] and TGF-β1 [81] to effect breast cancer and bladder cancer, respectively. miR-96 is also regulated by long non-coding RNAs (LncRNAs), such as lncRNA FGF14-AS2 [82] and lncRNA HOXC-AS3 [83], which target bind to it in GC and ovarian cancers (OC).

Generally speaking, miR-96 is abnormally highly expressed in a variety of cancers and plays a promoting role in tumorigenesis, mainly by targeting structural proteins and FOXO transcription factors. Moreover, its expression can be regulated by other non-coding RNAs. Importantly, our laboratory identified that miR-96-5p promotes tumorigenesis by directly targeting ZFAND5 in NSCLC, both in vitro and in vivo. miR-96 plays an important regulatory role in the occurrence and development of cancer. The study of the mechanism of miR-96 can provide new directions for cancer therapy. Summing up the research on miR-96 in cancer provides scientific support. Additionally, studying the miR-96-mediated pathway can offer fresh ideas for cancer detection and treatment.

### 3.4. Key Factors Targeted by miR-183 Cluster

miRNA clusters refer to miRNA and other miRNAs that are located in the same transcript, with the characteristics of clustering. The miR-183 cluster has been studied in the field of cancer and it has been discovered that it is abnormally expressed in high-incidence cancers such as lung cancer, CRC, PCa, GC, breast cancer, and bladder cancer. By binding with upstream and downstream target genes, it controls the invasion, proliferation, migration, and apoptosis of tumor cells. It has been demonstrated that the miR-183 cluster plays a crucial function in the development of cancer

At present, most studies have shown that miRNA cluster members have independent processing and regulatory pathways. However, there are few studies on the interaction between them. Through our result, it has been shown that this cluster of miRNAs cannot only regulate different genes to influence their occurrence and development, but also target the same genes, such as FOXO transcription factors, PTEN, TGFβ, and so on. The three members of the miR-183 cluster play both a uniform and specific regulatory role in cancer (Figure 3). These factors that can be co-targeted by miR-183 clusters are of particular interest to us.

FOXO transcription factors serve as the central regulators of cellular homeostasis and are tumor suppressors in human cancers [84]. The regulation and deregulation of FOXO transcription factors occurs primarily at the post-transcriptional and post-translational levels that are mediated by regulatory ncRNAs [85]. Our results suggest that the three members of the miR-183 cluster have a targeted regulatory relationship with these transcription factors. The human FOXO1 3′UTR contains a functional miR-183 site [86]. miR-182 is able to target FOXO family protein members in both lung and breast cancers [49,66]. MiR-96 is able to target FOXO family members in GC [76,77].

PTEN is a tumor suppressor that regulates growth and survival [87]. The three miR-183 cluster members have been found to control the expression of PTEN genes. [35,88,89]. The ability of miRNAs to control PTEN expression is used in the conventional method of regulation. Intriguingly, the PTEN pseudogene PTENP1 produces sense and antisense transcripts during transcription, demonstrating the posttranscriptional and transcriptional regulation of PTEN expression. PTENP1 plays a regulatory role in miRNAs, which allows PTEN to be released from miRNA-mediated repression [90,91]. PTENP1 regulates the expression of miRNAs, including miR-20 [92], miR-21 [93], miR-106b [94], and miR-19b [95]. The previous work made it clear that there is a relationship between PTEN and the miR-183 family members. Does PTENP1 possibly have the ability to control the miR-183 family as well? As there is not currently much relevant research, it will be crucial to develop this area of study in the future. It might even turn into a research hotspot.

Inflammatory illnesses are triggered by TGFβ signaling disruptions, which also encourages the development of tumors [96]. For example, miR-183 exerts a tumor-promoting role in OC and directly regulates biological functions through the TGFβ/Smad4 signaling pathway in OC cells [97]. MiR-183 inhibits the expression of TGFBR1 and the secretion of inflammatory factors in renal fibrosis and LN by directly targeting and blocking the TGF-β/Smad/TLR3 pathway [98]. Similarly, miR-182 and miR-96 are able to affect this TGFβ signaling [67,80].

Moreover, Shang et al. studied the production of clustered miRNAs and discovered that the adjacent miR-144 is necessary for the biogenesis of endogenous miR-451 [99]. As a result, we make speculations about whether the miR-183 cluster interacts during its formation and if it interacts with the onset and progression of illnesses. It is essential that we not only investigate how each miRNA regulates cancer growth and the incidence of this on an individual basis, but also pay more attention to whether there is a regulatory interaction between each miRNA in the cluster.

## 4. miR-183 Cluster and Other Noncoding RNAs

NcRNAs are closely related to the occurrence and development of malignant tumors [100]. Among them, the most studied are microRNAs, long non-coding RNAs, circular RNAs (circRNAs), and tRNA-derived fragments (tRFs). LncRNAs and circRNAs can act as competing endogenous RNAs (ceRNAs), containing miRNA-binding sites to function as miRNA sponges. LncRNAs have response elements (MREs) on mRNAs that are structurally and functionally similar to miRNAs and may act in a competitive manner [101,102].

In a variety of cancers, the lncRNA/circRNA-miRNA network plays an important role in regulating the expression of target genes and protein-coding genes. By constructing a GC lncRNA-miRNA-mRNA regulatory network and analyzing the network’s topology, it was found that lncRNA ADAMTS9-AS2, C20orf166-AS1, and miR-204 are key nodes [103]. In addition, another study established the same network, the members of which could be used as prognostic biomarkers for pancreatic cancer [104]. CircRNAs also function as cancer markers. Yang et al. [105] demonstrated that circ0005795 and circ0088088 form a network with miRNAs to regulate the occurrence and development of breast cancer. Another study found that five circRNAs could play key roles in cervical cancer as ceRNAs [106].

Studies on the miR-183 cluster’s regulation of lncRNAs and circRNAs have also been conducted recently (Table 4). In GC, LINCC00163 [107] and circ0000291 [108] can regulate AKAP12 and ITGB1 by targeting miR-183, affecting the tumor proliferation and migration. In PCa and CRC, lncCASC2 [109] competes with miR-183 to regulate Sprouty2, which, in turn, affects the drug sensitivity of the PCa cells. Circ0026344 [110] inhibits the metastasis of human CRC cells via miR-183. In addition, lncPCGEM1 [111] and circ0001776 [112] promote the occurrence and development of cervical cancer (CC) and inhibit the proliferation of endometrial cancer (EC) by targeting the miR-182/FBXW11 and miR-182/LRIG2 axes, respectively. LncTP53TG1 [113] promotes pancreatic ductal adenocarcinoma (PDAC) development by acting as a molecular sponge for miR-96.

Based on the above analysis, we can conclude that the expression level of the miR-183 cluster can be regulated by other ncRNAs and then affect the expression of downstream target genes to promote or inhibit the development of cancer. The members of ncRNAs play important regulatory roles in the occurrence and development of cancer. Studying the relationship between the miR-183 cluster and other types of ncRNAs is crucial for further exploring the function and regulatory mechanism of the miR-183 cluster in cancer.

## 5. Exosomal miR-183 Cluster in Cancer Cell Communication

In 1983, exosomes were first discovered in sheep reticulocytes. The research found that many biomolecules were present in exosomes such as proteins, mRNAs, non-coding RNAs, and so on [114]. The abundance of exosomes and their cell-to-cell presentation suggest that they make a difference in cell-to-cell communication and regulation. Researchers have listed that exosomes play important roles in a variety of biogenic processes, such as immune response, metabolism, the tumor microenvironment, cardiovascular diseases, and so on. Malignant tumors are characterized by high proliferation and migration. Therefore, it is speculated that exosomes are closely related to the development of tumor cells. Interestingly, the study of exosomes in the field of oncology has developed rapidly [115].

Research on the function of exosomes in cancer is developing quickly in comparison to other diseases. Exosomes have been associated with angiogenesis in the tumor microenvironment and extracellular matrix modification, both of which are essential for the growth, metastasis, and dissemination of malignancies. This may one day lead to a breakthrough in the treatment of cancer. Since exosomes are present in all bodily fluids and are produced by all cells, they are appealing for minimally invasive liquid biopsies, with a possibility for longitudinal sampling to track the development of the disease. For the delivery of certain miRNAs or small interfering RNA (siRNA) payloads, exosome engineering was created for CNS disorders and malignancies [115,116,117]. Based on their structure and function, they are promising as an early diagnostic marker for a variety of diseases and to serve as a vehicle for targeted drug delivery [118].

Serum and other bodily fluids contain miRNAs, which can act as disease indicators. Furthermore, secretory miRNAs, particularly those released by extracellular vesicles (EVs) like exosomes, may facilitate paracrine and endocrine communication between various tissues, regulating gene expression and remotely controlling cellular processes. When impacted, secretory miRNAs can cause disease, aging, and tissue malfunction [119,120].

For the miR-183 cluster, recent studies have shown that it exists in exosomes and has an impact on the development and progression of cancer. Figure 4 shows a schematic diagram of the miR-183 cluster in exosomes. By decreasing the expression of PPP1CA, the miR-183-5p that is produced in tumor cells, which is transported to macrophages via exosomes, stimulates the release of IL-1β, IL-6, and TNF-α from macrophages, limiting the anti-tumor immune response and sustaining the pre-tumor milieu [121]. Exosomal miR-183 can be secreted by non-tumor cells in the tumor microenvironment (TME) such as macrophages. In a study by Zhang et al., when miR-183-5p was downregulated, M2-TAM’s tumor-promotive effects on CC cells were reversed. The overexpression of miR-183-5p in M2-TAM exacerbated its promotive effects on CC cells [122].

In a study of the prostate, researchers demonstrated that the overexpression of miR-183 and exosomal miR-183 promotes cell proliferation, migration, and invasion [38]. The exosomal miR-96 in radiation-resistant NSCLC patients was significantly higher than that in the control group. It might be a non-invasive diagnostic and prognostic marker for radiation-resistant NSCLC [70]. Exosomal miR-182 can significantly promote the migration and invasion of GC cells by inhibiting RECK [123].

In addition, studies have shown that it has a stimulating effect on angiogenesis. Exosomal miR-182-5p promotes tumor angiogenesis by directly suppressing its targets such as Kruppel-like factor 2 and 4 and causing VEGFR to accumulate [124]. Exosomes made from CRC cells overexpressing miR-183-5p allow HMEC-1 cells to proliferate more rapidly, invade, and form tubes [125].

Studies on the miR-183 cluster in the context of cancer are somewhat scarce. However, according to the studies described above, the miR-183 cluster can exist in exosomes and influence the development of cancer. Furthermore, exosomes allow the exchange of goods between cancer cells and stromal cells in the tumor microenvironment. Therefore, it is of great scientific significance to analyze and summarize the research on the miR-183 cluster in exosomes to provide a preliminary basis for subsequent research.

## 6. Discussion and Perspectives

Human health is silently destroyed by cancer. Nowadays, surgery, chemotherapy, and radiation therapy are available as cancer treatments. Large tumors can be removed surgically, but surgery is invasive and does not remove metastatic cancer cells. Chemotherapy is a systemic treatment that destroys both primary and metastatic cells, but its drugs are not targeted enough, resulting in the destruction of healthy cells at the same time. Radiation therapy is utilized for locally sensitive tumors and many malignancies can be temporarily controlled with radiation therapy, but it is expensive and has many radiation complications. Recently, precision medicine has drawn more attention as a powerful strategy for fighting this deadly illness. We are also constantly investigating and developing the techniques of precise treatment in the fight against cancer. In tumorigenesis, making use of RNA and other potent characteristics that target genes is essential.

miRNAs, which have been proven as inhibitors of mRNA stability and translation, have a functional connection with the development of cancer. Since miRNAs are relatively stable in blood, serum miRNA levels can be used as a predictor of cancer and provide important information for diagnostics. Compared to the expression levels of mRNAs and proteins, miRNAs do not face the fate of being regulated and degraded, so their detection results are relatively more reliable. Most studies have indicated that miRNAs can be delivered in the form of nanoparticles, viral vectors, and lipid carriers [126,127]. At present, miR-34a is in many clinical studies. Hong et al. [128] proposed a phase 1 study of liposomal miR-34a in patients with advanced solid tumors, and Sharma et al. [129] reported the use of folic acid-targeted hybrid lipid polymer nano complexes for the co-delivery of DTX and miR-34a. In addition, miR-221 [130], miR-210 [131], and miR-193b-3p [132], etc., have been reported for their use in clinical drugs.

Exosomes harboring ncRNAs may be used as novel therapeutic targets in the treatment of cancers, such as preventing release to prevent tumor growth. Exosomes are found in a variety of bodily fluids, and because of their unique properties, they may be employed as a simple, slashing biomarker for identifying malignancies. Exosomes have significant promise for the diagnosis and prognosis of the illness. First, exosomes are tiny and prevalent in fluids, making them simple to gather and identify. Second, for a more accurate therapeutic agent delivery, exosomes can be genetically engineered and modified with tumor-targeting proteins, peptides, or antibodies [133,134,135]. Therefore, additional research into the functions and mechanisms of exosome-derived miRNAs is required to offer a novel approach to treating cancer [136].

In our study, the miR-183 cluster primarily affects tumor cell apoptosis, proliferation, migration, and invasion as a pro-oncogenic factor. It is abnormally expressed in several cancer forms, including NSCLC, CRC, PCa, and others. It may also create regulatory networks with other ncRNAs to either promote or prevent the onset and progression of cancer by targeting oncogenes. The miR-183 cluster also resides in exosomes, where it regulates the tumor microenvironment via exosomes. It plays a significant function in controlling the growth of cancer. Contradictory inferences are frequently made due to its wide modes of action in cancer. Furthermore, the miR-183 cluster contains a variety of downstream target genes that function in various ways. On the one hand, the three cluster members can control several target genes at the same time, such as PTEN, PDCD4, TGFβ, and FOXO transcription factors. On the other hand, they can target various impact factors individually. Additionally, the miR-183 cluster interacts with other ncRNAs, including lncRNAs and circRNAs, which can change its expression profile and impact its downstream target genes.

The study of the cancer regulatory mechanisms of the miR-183 cluster is still in its exploratory stage. There is currently not a very detailed regulatory action map. Does the miR-183 cluster interact with proteins and ncRNAs in a predictable manner, as regular as a mathematical model? Second, the miR-183 cluster’s ability to promote and inhibit cancer by affecting several biogenesis processes suggests that its mode of action is complicated. Due to its intricacy, it serves as a warning that more thorough, in-depth research in the future is still necessary. In addition, most of the research is at the cellular and molecular levels, and there have not been many experiments on animals. More reliable animal tests are required for a validation in clinical use. Therefore, we must continue our research and innovation. In general, understanding the miR-183 cluster is very important in how cancer develops. The miR-183 cluster is essential for comprehending cancer and its potential functional role offers novel cancer therapeutic options. The ability to use the miR-183 cluster as a clinical diagnostic tool for tumor identification opens up new possibilities for the targeted treatment of cancer.

## 7. Materials and Methods

The figures in the review were drawn using the Adobe Photoshop CC 2018 software and the tables were drawn using Microsoft Office Word. This review is based on previously conducted studies and does not contain any studies with human participants or animals performed by any of the authors.

## Figures and Tables

**Figure 1 cells-12-01315-f001:**
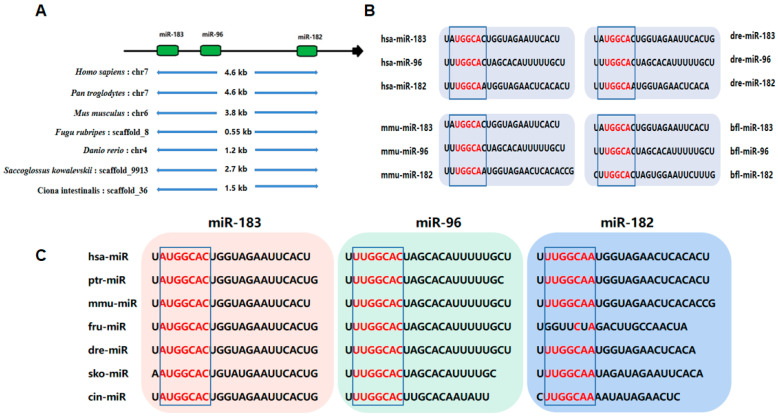
(**A**) The location of miR-183 clusters on the genomes of different species. (**B**) Sequence alignment of mature miR-183, miR-96, and miR-182 from different species. (**C**) Sequence alignment of mature miR-183, miR-96, and miR-182 from different species.

**Figure 2 cells-12-01315-f002:**
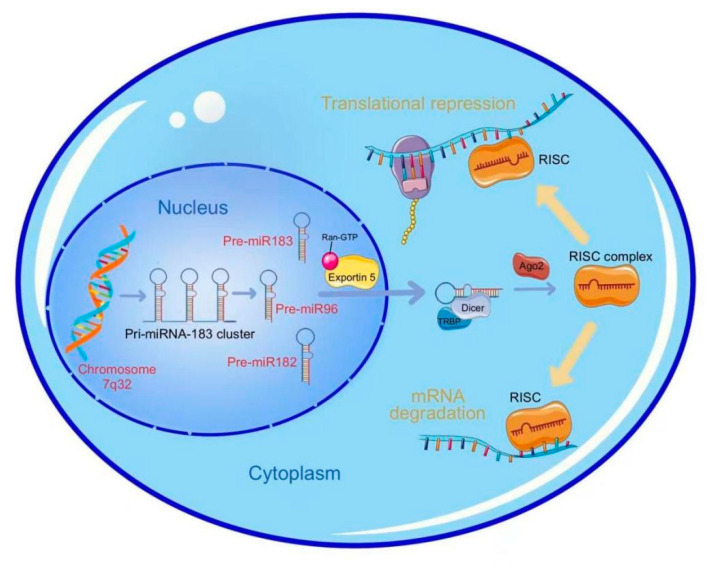
Biogenesis and biological role of miR-183 cluster.

**Figure 3 cells-12-01315-f003:**
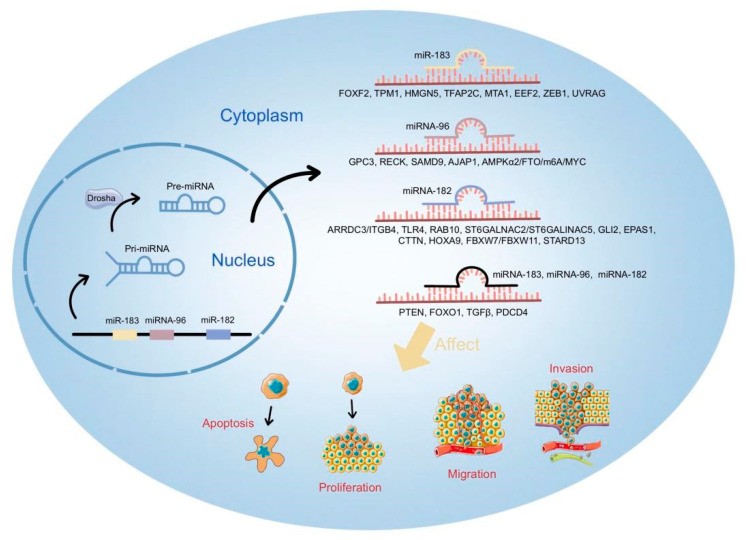
Regulatory map of miR-183 cluster in cancer.

**Figure 4 cells-12-01315-f004:**
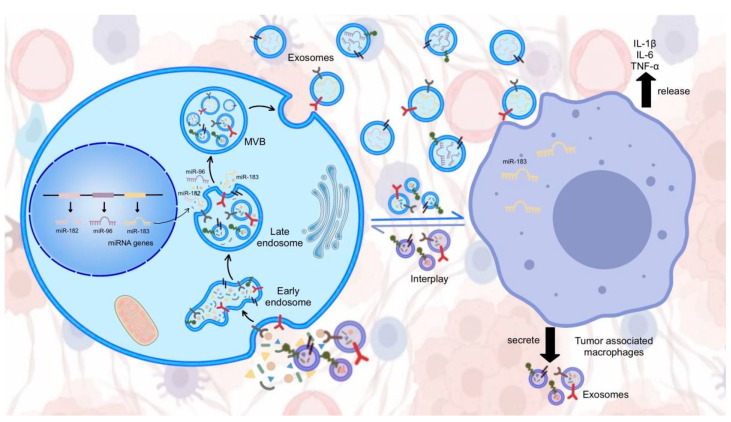
MiR-183 cluster in exosomes.

**Table 1 cells-12-01315-t001:** Functions of miR-183 in cancers.

Cancer	Regulation	Tissue No.	Cell Lines	Functions	Target Gene	Source
NSCLC	×	×	H157, 344SQ (mouse)	Inhibit EMT, migration, and invasion	Foxf2	[31]
NSCLC	Up	*n* = 8	NCL-H292, NCL-H838, MRC-5, WI-26 VA4	Promote occurrence and invasion	TFAP2C	[47]
NSCLC	Up	*n* = 76	A549, H226, BEAS-2B, H1395	Promote proliferation	×	[48]
NSCLC	Up	*n* = 55	A549, SPC-A-1, H1299	Promote migration and proliferation	PTEN	[35]
Lung cancer	Up	×	H1299	Promote EMT and radioresistance	ZEB1	[36]
NSCLC	Up	*n* = 194	SPC-A-1	Inhibit EMT, migration, and proliferation	MTA1	[44]
CRC	Up	*n* = 94	×	Invasion	×	[30]
CRC	Down	×	HCT116, HT29	Promote apoptosis	UVRAG	[37]
PCa	Up	*n* = 50	×	×	×	[33]
PCa	Up	×	LNCaP, PC-3	Promote migration, proliferation, and invasion	TPM1	[38]
PCa	Down	*n* = 12	Vcap, C4-2	Inhibit proliferation and migration. Promote apoptosis	HMGN5	[39]
GC	Down	*n* = 102	BGC823, MKN45	Inhibit proliferation and migration	EEF2	[40]
GC	Up	*n* = 24	GES-1, MKN-7, AGS, HGC-27	Promote migration and invasion	TPM1	[42]
GC	Down	×	MKN28	Inhibit invasion	UVRAG	[41]
HCC	Down	*n* = 10	HepG2	Inhibit proliferation and migration	MTA1	[45]
OS	Down	*n* = 40	×	Associated with progression and metastasis of osteosarcoma	×	[34]
OS	Down	*n* = 25	MG63	Inhibit the vitality, invasion and migration	MTA1	[46]

×: No relevant data or assays using pathological tissue or cell lines.

**Table 2 cells-12-01315-t002:** Functions of miR-182 in cancers.

Cancer	Regulation	Tissue	Cell Lines	Functions	Target Gene	Source
NSCLC	Up	×	NCI-H460, H1299	Enhance radioresistance	FOXO3	[49]
NSCLC	Up	×	NCI-H1975, NCI-H460, A549, 95-D	Promote EMT and migration	EPAS1	[50]
NSCLC	Up	*n* = 124	×	Oncogenic role	HOXA9	[51]
Lung cancer	Up	×	A549, PC-9	Promote migration, proliferation, and invasion	STARD13	[52]
NSCLC	Up	*n* = 11	H460	Promote proliferation	FBXW7, FBXW11	[53]
NSCLC	Up	×	A549	Chemoresistance	PDCD4	[54]
Lung cancer	Down	*n* = 27	NCI-H460, A549, CisR	Cisplatin resistance, tumorigenesis	GLI2	[55]
NSCLC	Down	*n* = 55	A549, H1299	Inhibit migration and invasion	CTTN	[56]
CRC	Up	*n* = 31	HCT8, LoVo	Promote chemoresistance	ST6GALNAC2	[6]
CRC	Up	*n* = 33	SW480, SW620	Tumorigenesis and invasion	ST6GALNAC2	[57]
CRC	Up	*n* = 159	DLD1, HCT15	Inhibit migration and proliferation	lncAGER	[58]
PCa	Up	*n* = 25	PC-3, Du145	Promote proliferation and invasion	ST6GALNAC5	[59]
Pca	Up	*n* = 82	MDA-Pca-2b, DU145, LNCaP	Invasion	PDCD4	[60]
Pca	Up	*n* = 147	DU145, LNCaP	Promote proliferation, migration, and invasion	FOXO1	[61]
PCa	Up	*n* = 65	HTB81, LNCaP	Promote proliferation, migration, and invasion and inhibit apoptosis	ARRDC3/ITGB4	[62]
PCa	Up	*n* = 25	PC-3, LNCaP	Promote proliferation, colony formation, migration, and invasion and inhibit apoptosis	Wnt/β-catenin	[63]
GC	Down	*n* = 148	×	×	×	[69]
GC	Up	×	GC-823, HGC-27	Inhibit migration and proliferation	Circ002059, MTSS1	[64]
GC	Down	×	AGS, HGC27	Regulate viability, autophagy, and apoptosis	Circ0001658, RAB10	[65]
Breast cancer	Up	*n* = 42	Jurkat	Raise the possibility of IL-17–producing Treg formation	CD3d, ITK, FOXO1, NFATs	[66]
Breast cancer	Up	×	Py8119, AT3, SCP28, Jurkat, CTLL2, PBMC	Promote breast cancer progression	TGFβ, TLR4	[67]
Bladder cancer	Down	*n* = 8	RT4, T24	Promote proliferation, migration, and invasion	Cofilin 1	[68]

×: No relevant data or assays using pathological tissue or cell lines.

**Table 3 cells-12-01315-t003:** Functions of miR-96 in cancers.

Cancer	Regulation	Tissue	Cell Lines	Functions	Target Gene	Source
NSCLC	Up	*n* = 52	×	Non-invasive diagnostic and prognostic marker of radioresistant NSCLC	×	[70]
NSCLC	Up	*n* = 57	A549, H460	Promote migration and invasion	GPC3	[71]
NSCLC	Up	*n* = 5	H358, H23	Induce cisplatin chemoresistance	SAMD9	[72]
CRC	Up	*n* = 60	SW480, SW620, HCT-8	Promote proliferation and inhibit apoptosis	AMPKα2/FTO/m6A/MYC	[73]
CRC	×	×	HCT-116	Promote invasion	RECK	[74]
PCa	Up	×	PC-3	Promote migration	FOXO1, FOXO3a	[79]
PCa	Up	*n* = 13	PcaP, PC3	Promote proliferation	FOXO1	[75]
PCa	Up	×	DU145, LNCaP	Inhibit proliferation	LncRNA FGF14-AS2, AJAP1	[82]
GC	Up	*n* = 70	MKN45, SGC7901	Promote proliferation	FOXO3	[76]
GC	Up	×	SGC7901	Promote proliferation, migration, and invasion	FOXO1	[77]
brest cancer	Up	*n* = 44	MCF-7, MDA-MB-231	Promote migration	MTSS1	[80]
HCC	Up	*n* = 60	HepG2	Inhibit proliferation, migration, and invasion	FOXO1, AKT/GSK-3β/β-catenin	[78]
bladder cancer	×	×	HT1376	Regulate migration and invasion	TGF-β1, FOXO1	[81]
OC	Up	*n* = 62	Caov-3, OVCAR3	Promote migration	LncRNA HOXC-AS3	[83]

×: No relevant data or assays using pathological tissue or cell lines.

**Table 4 cells-12-01315-t004:** Functions of miR-183 with other noncoding RNAs in cancers.

Cancer	miRNA	ceRNA	Functions	Source
GC	miR-183	LINCC00163	Inhibit GC via miR-183/AKAP12 axis	[107]
GC	miR-183	Circ0000291	Promote GC via miR-183/ITGB1 axis	[108]
PCa	miR-183	LncCASC2	Compete with miR-183 to rescue the expression of SPRY2 inhibit cancer	[109]
CRC	miR-183	Circ0026344	Inhibit CRC via miR-183	[110]
CC	miR-182	LncPCGEM1	Activate the NF-κB and β-catenin/TCF signaling pathways	[111]
EC	miR-182	Circ0001776	Inhibit EC via miR-182/LRIG2 axis	[112]
PDAC	miR-96	LncTP53TG1	Promote tumor via miR-96/KRAS axis	[113]

## Data Availability

All data included in this manuscript are from published scientific manuscripts and available on Google Scholar, Web of Science and PubMed (NIH).

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
