# Peer review of "The miR-183 Cluster: Biogenesis, Functions, and Cell Communication via Exosomes in Cancer"

_cells, 2023, doi:10.3390/cells12091315_

Round 1
Reviewer 1 Report (New Reviewer)
In this manuscript Authors provided a review on the role of miR-183 cluster in tumorigenesis. In the first part of the review Authors described the role of each member of the cluster in cancer biology, mainly describing their direct targets, and then they focused on the targets (and associated processes) regulated by more than one miRNA of the cluster. In the second part, they outlined the existence of several regulatory axes involving miR-183 cluster members and other non-coding RNA. Finally, they addressed the role of exosomal miR-183 cluster in cell-to cell communication.
Though some reviews regarding this topic already exist in the literature, the novelty of this work could be the description of the role of the cluster in both i) cell-to-cell communication and ii) regulatory circuitries involving other non-coding RNA.
Although the scope of the review is clear, the manuscript has some issues and some drawbacks.
1) The text needs to be thoroughly revised for proper English usage, since it is poorly written in many, if not all, sections. A lot of sections are confusing and might lead to a misinterpretation of what is being said.
2) Page 2 line 80. It is generally thought that miRNAs regulate their targets either by translation inhibition or by target destabilization (not target cleavage) as Authors correctly state in the discussion.
3) Paragraphs 3.1, 3.2 and 3.3. The Authors comments in the concluding part of the paragraphs are essentially the same for all three paragraphs. Authors may consider to better articulate the discussion highlighting the common elements and the peculiarities of the three miRNAs activities. Moreover, Authors should comment the fact that for some cancer type controversial data exist regarding the possible role (tumor promoting or tumor suppressor) of the same miRNA of the clusters.
4) Page 4. Authors described the target that are regulated by more than one miRNA of the cluster. Firstly, I think that this part should be paragraph 3.4 and should have an appropriate title. Secondly, according to the text and especially Figure 3, some targets are directly regulated by all three miRNAs of the cluster. Is this the case? Because I could not find data regarding miR-183 direct regulation of Foxo3a or miR-182/miR-96 direct regulation of LRP6. I think that a table in which the type of regulation (direct or indirect) is specified for each miRNA could avoid misinterpretation of Figure 3. Moreover, Authors should specify this issue in the legend of Figure 3. Thirdly, some sentences need a literature reference (“MiR-182 and miR-96 were able to target FOXO family proteins members in both lung and breast cancers”; “The expression of the PTEN genes was discovered to be regulated by three cluster members”; “Similarly, miR-182 and miR-96 were able to affect TGFB signaling”). Finally, the involvement of PTEN/PTENP1 in miRNA cluster regulation (lines 240-245) and the Authors considerations regarding the 183 cluster processing considering the work of Shang et al. (lines 252-260) are not comprehensible to me. Authors should rephrase the concepts.
5) Chapter 4. The involvement of exosomal miR-183 cluster in cell-to-cell communication is the novelty of the review. Therefore, Authors should consider expanding this section. There are some papers that Authors did not consider (PMID 34249713, 32364530, 32366676) related to the exosomal miRNA cluster members secreted by non-tumor cells of the tumor microenvironment (TME) such as macrophages. Authors should consider these papers and try to describe a more exhaustive view of the activity that miRNA 183 cluster plays in cell-to-cell communication in the TME, modifying Figure 4 accordingly.
Author Response
Please see the attachment.

Reviewer 2 Report (New Reviewer)
microRNAs have been associated with almost every normal cell function, including proliferation, differentiation and apoptosis. Why was the natural role of this microRNA not indicated?
This microRNA may be present in the circulatory system outside the exosomes. Has the difference between them been studied in depth? What is the importance of being present inside the exosome in the field of diagnosis.
Line 48: Report not repot
Line 55: authors: not the right word to use
The figures are not explained enough. The method of drawing figures and the software used must be clarified.
The authors superficially cover studies of the exosome in this review study
Author Response
Please see the attachment.

Reviewer 3 Report (New Reviewer)
In their review "MiR-183 cluster : biogenesis, functions, cell communication via exosome in tumorigenesis", Shuhui Li et al comprehensively summarize the role of this miRNA cluster in various cancer entities.
While I generally support publication of this review, I would suggest a few minor adaptations to further improve the manuscript:
1) In the title, I think it should exosomes instead of exosome and maybe cancer instead of tumorigenesis. Furthermore, I think the relevance stated in the abstract is too far fetched (therapeutic option)
2) Line 30: 'alterations in several genomes' makes no sense to me; and the following sentence is a bit disconnected
3) Line 48: 'reports'
4) Line 52: I am unsure if 'exosomes were used to bundle cluster members' is the right term here
5) line 59 and others: reconsider your choice of tenses. the miR cluster 'was' not located, but it 'is'
6) Line 63: add reference!
7) Line 68: Figure 1 is not actually related to evolutionary conservations. And it would be nice to add the 2D structure of the miR-cluster transcript to the figure as well
8) miR-183-5p has been shown to be expressed as different 5'isomiRs with different functions. The authors should add this aspect to their article as well.
9) in the tables summarizing miR roles in different studies, it would be nice to split them visually into studies supporting pro-tumor and anti-tumor phenotypes. same holds true also for table 4.
10) Line 209ff: should this paragraph have its own headline? And I do not understand the point the authors want to make in line 217ff
11) Line 323: The PPP2CA should be brought into contextwith exosomes or a phenotype
12) Line 326: I think this is not a proper sentence. And in line 329: how can exosomal miRNAs inhibit antibodies?
13) Line 339 and 404: It's unclear how studying exosomes is expected to result in a breakthrough for cancer treatment. Please elaborate or tone down.
14) Line 343ff: The authors make it sound line cancer is always recurring and refractory to the treatment. They should phrase this a bit more realistically.
15) In Discussion/Outlook, the authors should reduce a bit the part talking about other miRs or miRs in general. It is distracting.
16) Line 394: what 'both proteins' are the authors referring to?
Lastly, the authors should carefully read the text again to spot typos and grammar mistakes or ' isn't ' or similar phrases.
Round 2
Reviewer 1 Report (New Reviewer)
The Authors have addressed almost all my concerns. I think the manuscript still needs a further revision in English; there are many typos and sentences not written properly.
Author Response
Please see the attachment.

This manuscript is a resubmission of an earlier submission. The following is a list of the peer review reports and author responses from that submission.
Round 1
Reviewer 1 Report
This review paper entitled “MiR-183 cluster: biogenesis, functions, cell communication via exosome in tumorigenesis” by Shuhui Li et al. discusses the bioprocesses, functions, and cellular locations of the microRNA183 cluster containing three members, miR-183, -96, and -182. The functional properties of each individual member are discussed, and these discussions link this microRNA clusters to cancers, which is informative. However, the present form of this paper seemed to be a stack of materials, which is not easy to follow. This should be significantly revised. Specifically.
1) Please organize the materials following a logical stream and pay attention to the logic between the statements. For example, what is the logical link between “main methods of cancer treatment” and ncRNAs from lines 34 to 38?
2) Discussions can be broad or deep.
Broad: For example, recent big data studies have revealed that noncoding RNAs (ncRNAs) endogenously rule the cancerous regulatory realm and that there is a distinctive functional regime of endogenous lncRNAs in the human genome. MicroRNA-183 cluster has also been found in multiple cancer types. It would be interesting to collect data to demonstrate its richness across cancers and its functional systems. Overall, this data indicates its importance in cancer.
Deep: One example is PTEN, which is regulated by ncRNAs as discussed in your paper. Conventional versions of PTEN regulated by ncRNAs such as PTENP1 (PTEN pseudogene) and microRNAs have been updated by recent big data studies, in which it has been expanded to a network module containing several novel PTENP1 interacting partners in the cancer realm, such as PTENP1_AS, RP11-181C21.4, PTENP1-MEMO1P1, and RP11-384P7.7. Can you provide data to show how microRNA-183 members interact with this module or its module components?
3) The microRNA interaction network described in this paper should be in a systems network form.
Reviewer 2 Report
In the manuscript entitled ‘’MiR-183 cluster : biogenesis, functions, cell communication via exosome in tumorigenesis’’, Li et al. reviewed the regulatory role of MiR-183 in the carcinogenesis via exosomes. I think that the structure of this review is unclear and disorganized. In general this review is poorly written and the sentences often lack of consequentiality. English is very hard to understand and overall the entire text is not fluent. The figures are not well prepared. For these reasons, I think that the review, in its current version, does not meet the standard for this journal.